# Health system actors' perspectives of prescribing practices in public health facilities in Eswatini: A Qualitative Study

Nondumiso B. Q. Ncube[1]*, Lucia Knight[1], Hazel Anne Bradley[1],
Helen Schneider[1], Richard Laing[1,2]

1 Department of Community and Health Sciences, School of Public Health, University of the Western Cape, Bellville, Cape Town, South Africa, 2 Boston University School of Public Health, Boston, Massachusetts, United States of America

☯ These authors contributed equally to this work.
* nncube@uwc.ac.za

**Data Availability Statement:** All relevant data are within the manuscript and its Supporting Information files.

## Abstract

### Background

Rational medicines use (RMU) is the prescribing/dispensing of good quality medicines to meet individual patient's clinical needs. Policy-makers, managers and frontline providers play critical roles in safeguarding medicine usage thus ensuring their rational use. This study investigated perspectives of key health system actors on prescribing practices and factors influencing these in Eswatini. Public sector healthcare service delivery is through health facilities (public sector, not-for-profit faith-based, industrial) and community-based care.

### Methods

A qualitative, exploratory study using semi-structured in-depth interviews with seven policy-makers and managers, and 32 facility-based actors was conducted. Drawing on Social Practice Theory, material (health system context), competence (provider) and cultural (patient and provider) factors influencing prescribing practices were explored.

### Results

Participants were aged between 21-57years, had been practicing for 1–30 years, and were a mix of doctors, nurses, pharmacists and pharmacy-technicians. Factors contributing to irrational medicines use included: poor use of treatment guidelines, lack of RMU policies, poorly-functioning pharmaceutical and therapeutics committees, stock-outs of medicines, lack of pharmacy personnel in primary healthcare facilities, and restrictions of medicines by level of care. Provider-related factors included: knowledge, experience and practice ethic, symptomatic prescribing, high patient numbers. Patient-related factors included late presentation, language, and the need to be prescribed many medicines.

### Conclusion

In Eswatini, prescribing practices are influenced by the interaction of factors (health system, provider and patient) that span levels (facility, region, and policy-making) of the health

**Funding:** NBQN received support towards doing a PhD at the School of Public through an agreement between the Institute of Tropical Medicine and the School of Public Health, University of the Western Cape. The funders had no role in study design, data collection and analyses, decision to publish, or preparation of the manuscript.

**Competing interests:** The authors have declared that no competing interests exist.

system. Promoting RMU thus goes beyond the availability of guidelines and provider training and requires concerted efforts of multiple stakeholders.

## Introduction

Rational use of medicines is crucial to well-functioning health systems. The World Health Organization (WHO) considers that medicines are used rationally when individual patients are prescribed and dispensed correct medicines, of good quality, in appropriate doses to meet their clinical needs, at a minimal cost to them and their community, and for an appropriate duration of treatment [1]. Health system actors, including policymakers, essential medicines lists/formulary committee members, prescribers, and pharmacists, play critical roles in safeguarding the use of medicines, from procurement until they reach the end-user [2]. In healthcare facilities, frontline providers (prescribers and pharmacy personnel) ensure that medicines are ordered from the central medical stores (CMS), stored correctly, and used rationally.

Prescribing practices, also referred to as prescribing behaviours, are key to understanding patterns of medicines use. This research draws on the Social Practice Theory (SPT) to study prescribing practices in public sector and not-for-profit faith-based facilities in the Kingdom of Eswatini (formerly Swaziland, and hereafter referred to as Eswatini). The SPT seeks to explore the relationship between available resources, the nature of practice, and contexts within which individuals function; and suggests that practices, and changes in practice, are shaped not only by individual competence but also by the social (shared meanings) and material (resources) contexts [3, 4] as illustrated in Fig 1.

The SPT is a socially oriented approach to analysing behaviour that is useful in gaining insights into the processes and structures that generate behaviour [5]. It is based on the understanding that individualist approaches have not been effective in creating expected changes as they tend to ignore the wider context of practice and function as specific "corrective" action. Socially orientated approaches, on the other hand, help to develop new strategies for changing behaviour involving multiple stakeholders, and may require all these stakeholders to do their daily activities in a different manner [5].

Literature on prescribing behaviour highlights a range of provider, patient and health system factors that can be mapped onto the domains of the SPT (Table 1).

In Eswatini, guides to prescribing include The Standard Treatment Guidelines and Essential Medicines List (STG/EML) of Common Medical Conditions in the Kingdom of Swaziland, published in 2012 [17]. The STG gives guidance on how conditions should be managed pharmacologically and non-pharmacologically as well as the level of care at which the pharmacological treatment can be available. The EML gives a list of medicines (including its strength, dosage form) approved for use in Eswatini; the vital, essential, non-essential (VEN) allocation of the medicine; and the level of care (A = all health facilities; B = Health centers; C = Hospitals; S = prescribed by specialist doctors) at which the medicine can be used. Though literature shows that availability of guides such as essential medicines lists may improve prescribing practices [18], there is no evidence on whether the Eswatini STG/EML has influenced the use of medicines in the country. At the time of data collection, an implementing partner (Management Sciences for Health/ Systems for Improved Access to Pharmaceuticals and Services (MSH/SIAPS)) was providing support for strengthening pharmaceutical systems in the country. A survey conducted by MSH/SIAPS in 2013 revealed that there was no continuous professional development training available on rational medicines use (RMU), although small-scale

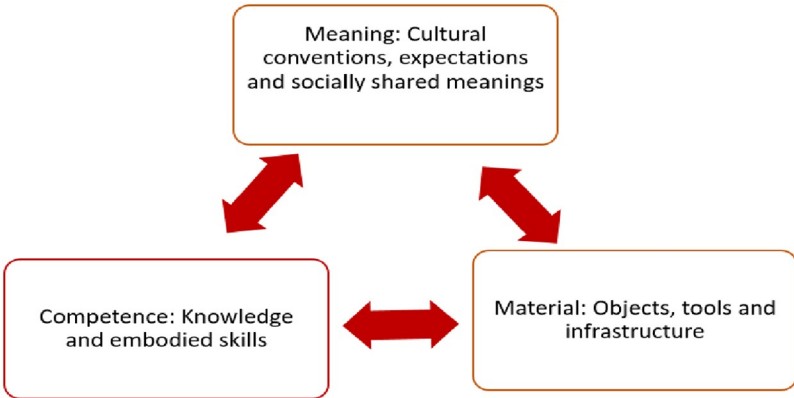

**Fig 1. The social practice theory.**

facility-based RMU training was taking place [19]. This is in line with statements in the National Pharmaceutical Policy (NPP) (2011), which highlight that the country does not have mechanisms for monitoring medicine use by health workers and the public, mainly due to lack of the necessary tools, staff and resources [20]. The NPP further recommends that existing prescribing and dispensing practices need to be rationalized and streamlined through the development of various RMU tools and staff training.

Studies on prescribing practices have mainly been quantitative using the WHO/INRUD prescribing indicators, which assess the following: average number of medicines per patient encounter, percentage of medicines prescribed by generic name, percentage of prescriptions with one or more antibiotic prescribed, percentage of prescriptions with one or more injection prescribed, and percentage of medicines prescribed from the essential medicines list or formulary [21, 22]. Literature on reasons behind certain prescribing practices using qualitative approaches is scant. Minimal work has been done on prescribing practices in Eswatini and this study seeks to fill this gap.

The aim of this study was to explore policy-maker, manager and frontline healthcare providers' perspectives on existing prescribing practices and the factors influencing these in Eswatini. Specifically, it investigated the perspectives of policymakers in medicine management (senior pharmaceutical staff in the Ministry of Health (MoH) including CMS and regional level staff), implementing partners supporting pharmaceutical services, and frontline

**Table 1. Factors affecting prescribing behaviours and how they link to the SPT.**

| Type of Factor and SPT Element | Factors |
| --- | --- |
| Provider (Competence) | Practitioner's experience and level of education, inadequate training of healthcare providers, visits by pharmaceutical sales representative [6–8], physician's age and gender [9, 10], prescribers' perception that the patient wants certain medicines and fear that not giving antibiotics will lead to patients having medical complications [11, 12], fear that prescribers will lose patients if they do not give in to their demands [13]. |
| Patient-provider interaction (shared meaning) | Patient expectations and demands [13]. |
| Health system context (Material) | Health systems factors such as lack of continuous professional development, lack of structures that provide updated and evidence-based information on medicines in current use, patient load, and influences of pharmaceutical representatives [6–8], socio-economic characteristics of the environment in which practice is conducted, health demand [9], availability of materials to work with [14, 15], and availability of funds to procure medicines [16]. |

healthcare providers (prescribers (doctors and nurses), pharmacists and pharmacy technicians in the MoH) on prescribing practices in Eswatini.

## Methods

### Study setting

Eswatini is a landlocked country in Southern Africa sharing its borders with South Africa and Mozambique. The country has an estimated surface area of 17,364 square kilometres and a population of about 1.2 million [23]. Most of the population resides in rural areas with only 22% in urban areas [24]. The country is divided into four administrative regions: Hhohho, Manzini, Shiselweni and Lubombo. In the health sector, healthcare services are delivered in both private and public sectors. The public sector provides services through public, not-for-profit faith-based, and industrial health facilities (clinics and outreaches, public health units, health centres and hospitals), and community based care (faith-based healthcare providers, rural health motivators and volunteers) [25]. While clinics only provide primary healthcare services, health centres and hospitals provide both primary and secondary healthcare services. Clinics are manned by nurses who are responsible for prescribing and dispensing medicines; health centers have nurses and doctors responsible for prescribing while pharmacy technicians dispense medicines; and hospitals have nurses and doctors responsible for prescribing while pharmacists and pharmacy technicians dispense medicines. Previously, the government did not have posts for pharmacy assistants. Posts for pharmacy assistants to be employed in all levels of care, particularly clinics where there were no pharmacy personnel as part of human resource, were approved in 2017 [26].

The MoH is responsible for ensuring that national health-related administrative and executive functions are performed adequately in the country. It also provides guidance on essential health care package delivery to all levels of healthcare countrywide. The MoH decentralises its activities to regional health offices (RHO) in the four regions, and the RHOs are responsible for the implementation of national health plans and policies. At the regional level, the regional health management team (RHMT) provides technical leadership in the implementation processes. The RHMT comprises of the regional health administrator (overall in-charge), a senior matron from the regional health office, senior medical officers and matrons from hospitals and health centres, clinic sisters (senior nurses based in the different clinics in the country), a pharmacist, and other health professionals stationed at the RHO depending on their availability.

### Conceptual framework

To study rational medicine use practices among policy makers at the national department of health (NDoH) and frontline medicine managers (prescribers (doctors and nurses), and pharmacy personnel (pharmacists, pharmacy technicians and pharmacy assistants)) in Eswatini, this study used a theory informed approach.

The bigger study was conducted in three phases; this paper focuses on the qualitative aspect of Phase I using the SPT. Phase I was a situational analysis to establish practices and processes in place regarding RMU in the country. Elements of the SPT were used to develop the data collection tools for assessing medicine prescribing practices among health actors in Eswatini.

### Study design and sampling

A qualitative exploratory study design was used to conduct this study. A qualitative exploratory study is defined as a study that allows researchers to investigate topics that have received

minimal attention in the past and are not well defined [27]. Further, qualitative exploratory study designs allow study participants to contribute information towards building knowledge in the area under study [28].

This research is part of a baseline survey conducted for a larger study aimed at improving RMU in Eswatini. Facilities were randomly sampled for the bigger study. Sampling for facilities has been thoroughly investigated by the INRUD group which recommends a minimum of 20 facilities to allow researchers to draw meaningful conclusions[29]. A sampling frame of 325 public and faith-based facilities that receive essential medicines from the central medical stores (CMS) was obtained from CMS. Faith-based facilities are public sector facilities that also receive support from faith-based organizations such as churches. The CMS codes facilities by region and these codes were used to assign facilities to the four regions (Hhohho, Manzini, Lubombo, and Shiselweni) in the country. Specialized facilities (i.e. national referral hospital, psychiatric hospital, tuberculosis (TB) hospital and those facilities governed by other bodies such as the police services and the army) were excluded from the sampling frame, leaving 286 eligible facilities. Clinics only provide primary healthcare and are staffed with nurses. Health centres and hospitals provide primary and secondary healthcare and are staffed with nurses, doctors, pharmacists (hospitals only), and pharmacy technicians (hospitals and health centers). A patient can choose to attend at any level of care they want to go to regardless of presenting condition. Very ill patients are referred from clinics to health centres or hospital depending on the proximity.

From the 286 facilities, one hospital per region (4) and all health centres in the country (5) were purposively included leaving 277 clinics to sample from. To sample clinics, a random sequence was used to select five clinics per region. To allow for non-response, the sample of clinics was inflated by 20% per region (making six clinics per region). A total of 24 clinics (six per region) were included in the sample, making the overall sample size 33. One facility (a clinic) in the Shiselweni region was closed on the day the principal investigator (PI) went to collect data. Efforts were made to find out on the functioning of this facility and reasons that were beyond the research team were given. Since the sample of clinics had been inflated by 20%, the research team decided to drop this facility leaving the total number of facilities at 32. Table 2 provides characteristics of respondents by cadre and level of care.

Key informants (7) were purposively selected based on their involvement in RMU policy decisions, while frontline managers (32) were from the different levels of care available in the country, as randomly sampled for the bigger study. Frontline managers were recruited into the study based on their presence at work on the day of data collection and their willingness to participate in the study. Only frontline managers involved in medicine use such as prescribers (doctors and nurses) and dispensers (pharmacists and pharmacy technicians) were approached for recruitment into the study.

## Data collection

A semi-structured interview guide informed by the SPT and covering themes such as knowledge of RMU, prescribing practices, enablers and barriers to the rational use of medicine, and interventions available to promote RMU was used to conduct one-on-one interviews with key informants. A similar interview guide, but focusing on RMU activities and functioning in facilities, was used to conduct one-on-one interviews with frontline healthcare providers. Interviews were conducted between April and September 2017. Semi-structured in-depth interviews were used to collect data on medicine use practices in public sector and faith-based facilities in Eswatini. Semi-structured in-depth interviews were used as they employ the use of an interview guide with open ended questions to allow a range of responses from participants in a

**Table 2. Study participants' characteristics.**

| Characteristics | Cadre (Number and Region) |
| --- | --- |
| Key Informants (N = 7) | |
| Ministry of Health (National level of care) | Pharmacist (1: Hhohho region) |
| Implementing Partner | Pharmacist (1: Hhohho region) |
| Ministry of Health (Central level of care) | Pharmacist (1) |
| Ministry of Health (Regional level of care) | Nursing Matron (4: 1 from each region) |
| Frontline Managers (N = 32) | |
| Ministry of Health (Secondary level of care) | Medical Officer (1: Lubombo region) Pharmacist (2: 1 from Hhohho and 1 from Shiselweni regions) Pharmacy Technicians (6: 4 from health centres [2 from Hhohho; 2 from Shiselweni] and 2 from hospitals [1 from Shiselweni and 1 from Manzini]) |
| Ministry of Health (Primary level of care) | Nurse (23: 6 from Manzini; 5 from Shiselweni; 5 from Lubombo; 7 from Hhohho)* |

*one clinic from the Hhohho region was wrongly coded as a facility in the Lubombo region by the central medical stores—it was analyzed under the Hhohho region. One clinic in the Shiselweni region was closed during data collection.

comprehensive and systematic manner to obtain the desired information [30]. Interview guides were piloted on healthcare professionals from facilities that were not in the sample (for frontline managers) and the CMS (for KIs). All interviews were conducted in English and the interviews lasted between 20 and 60 minutes. No repeat interviews were carried out.

The principal investigator (NBQN, a female PhD student) approached KIs and requested them to participate in the study after giving verbal and written information on the study. In facilities, the principal investigator (PI) introduced herself to management using the permission letter from the MoH and management guided on participants to be approached. NBQN conducted key informant interviews in offices in the MoH headquarters, implementing partner's office, and CMS; while a quiet office with minimal disturbance was used in health facilities. Interviews were audio recorded (all except one non-consenting participant), and were supplemented by field notes using a reflexive journal by NBQN. Some of the participants were people that NBQN previously worked with while practicing as a pharmacist in Eswatini, while others were not known to the PI. For frontline managers, data saturation, a phenomenon achieved when the data gathered no longer brings out new information [31], was achieved by the 15th interview. However, since the PI was collecting quantitative data from 32 facilities, she continued conducting interviews in all facilities. A research assistant transcribed the interviews and saved transcripts onto a password-protected laptop. Some participants (3 key informants) were interested to review transcripts and they were given the opportunity to do so (member-checking) before finalization of the transcripts. Frontline managers were not keen on reviewing their transcripts.

## Ethical considerations

Ethics approval to conduct the study was obtained from the Biomedical Science Research Ethics Committee of the University of the Western Cape (Ref: BM/16/4/2) and the National Health Research Review Board in Eswatini. Permission to access healthcare facilities was granted by the office of the Deputy Director Pharmaceutical Services in the MoH. Consenting participants were given verbal and written information (information leaflet) on the study and

were assured that they were free to refuse participation with no repercussions. No incentives were provided for being part of this study. Written consent was obtained from all participants.

## Data management

Data were anonymized and securely stored on a password protected computer. A research assistant transcribed the audio recordings verbatim. The first author (NBQN) re-checked all the transcripts for completeness and correctness. Two of the authors (NBQN, LK) read through the transcripts thoroughly for familiarization. Two transcripts were randomly selected from the lot and independently coded by NBQN and LK on Atlas.ti. Notes and memos were made from the selected transcripts to inductively generate the initial codes. NBQN and LK discussed their independent codes identifying similarities and differences through a discursive process. Using the initial codes, NBQN and LK developed a codebook. Codes with similar ideas were clustered to form sub-themes. Sub-themes with identical concepts were grouped deductively into final themes provided by the SPT framework. NBQN used the codebook to analyse the remaining transcripts. The reorganizing of codes was achieved in a discursive process with all the authors.

The Framework Method [32], which is part of the thematic analysis family [33] was used to analyze the data. Thematic analysis is defined as a method in qualitative research that is used to identify, analyse and report themes or patterns within data; prescribing patterns in this study [33]. Using the Framework Method, we applied a hybrid of inductive and deductive thematic analysis. The initial coding was done inductively and the themes emanating from the coding were then classified deductively using the SPT framework. Thematic analysis was chosen for this study because it is a flexible method that allows researchers to produce a rich and detailed account of data [33].

## Results

Study participants consisted of a mix of doctors, nurses, pharmacists and pharmacy technicians. Their ages ranged from 21–57 years, and they had been practicing for periods ranging from 1–30 years (median = 12.7 years).

Key themes that emerged from the data included contextual, healthcare provider, and patient oriented factors. Themes were organized around the SPT elements (material, meaning and competence), factors identified in literature to affect prescribing practices, and factors emerging from the data.

### Health system context factors (material)

This section will highlight factors that participants reported to affect prescribing practices in line with how literature suggests that the context within which practice happens affect rational use of medicines. Availability of policy to guide RMU practices, availability and use of the STG/EML, availability of essential medicines, lack of pharmacy personnel in primary levels of care, restrictions of medicines by level of care, and active Pharmaceutical and Therapeutics Committees (PTCs) in hospitals and health centres were reported as health system factors (material) that affect prescribing and rational use of medicines.

**Availability of policies to guide RMU practices.** Frontline healthcare providers mentioned that the country had no policies to guide RMU, and this made it difficult for healthcare professionals to practice RMU without guiding policies. One frontline healthcare provider mentioned that due to lack of policies there was no guidance on who in the nursing cadre can and cannot prescribe, and this resulted in irrational prescribing.

**Availability and use of the STG/EML.** Information from key informants highlighted that public sector prescribing practices were influenced by the current STG/EML; with the STG having been mainly targeted at primary health care (though they can still be used at secondary and tertiary levels of healthcare), and the EML covering all levels of care. Until the latest guidelines published in 2012, Eswatini did not have country-specific guidelines. The current STG was a starting point to mainly guide clinics which provide primary healthcare and are only managed by nurses with no doctors and pharmacists; though the EML is comprehensive and includes all medicines available in the country at the different levels of care. The Ministry of Health had planned to then develop a more comprehensive guideline at a later stage. However, key informants highlighted that targeting the STG at primary level of care resulted in secondary and tertiary level facilities prescribing outside the guidelines.

> "At tertiary level they say our STG is skewed a lot towards primary health care and doesn't provide for their guidance. And of course it is a little bit true... there is some truth to it because you know at tertiary level we can do with a little bit of a revision and we do both tertiary level and primary health care level [guideline needs to be revised for both primary and tertiary level], because at primary health care level we know that we do not have medical practitioners so we thought those are the ones that needed the guidance the most that's why we started with them (KI_P_N_H)."

Key informants also reported that though the STGs/EML were outdated as they were published in 2012, they were still in use at the time of data collection for this study in 2017. However, KIs reported that availability of the STG/EML positively influenced rational use of medicine as some facilities used the STG/EML to develop their own (facility-specific) formularies. Though the STG/EML was seen to promote rational use of medicines, key informants reported that adherence to these guidelines seemed poor.

> "We have put in place some contingency measures to try and guide or to try and coerce people to use medicines rationally by putting in place standard treatment guidelines. But when we follow with our facilities, we have realised that it is not used as such because some of them when you visit facilities, they have to look for the STG within their cabinets and stuff, yet you'd expect it on the desk where they are working with it (KI_P_N_H)."

A reason that was cited by key informants to possibly result in poor adherence to guidelines was that frontline healthcare providers were not comfortable to use the guidelines in front of patients.

> "Because people think that [pause] I don't know it has not been proven so I want to believe it's a perception of the health care workers, because they tend to think that if they flip through their STG the patient will think they do not know, each time they are treating them they have to be checking but maybe with a mobile app on their smart phone the fears will be allayed (KI_P_IP_H)."

Key informants further reported that poor adherence to the STG/EML resulted in overuse of antibiotics which will in the near future contribute to the global problem of antimicrobial resistance. On the contrary, frontline healthcare providers validated overuse of antibiotics by reporting that most of the conditions that patients presented with required management with antibiotics.

**Restrictions of medicines by level of care.** Frontline healthcare providers highlighted the difficulty for them to adhere to the STGs due to restrictions on availability of medicines at certain levels of care. An example cited was the unavailability of ceftriaxone and azithromycin at primary healthcare levels yet the latest guidelines for managing sexually transmitted infections (STIs), published in June 2018, recommend these medicines as first line therapy and for them to be available at primary healthcare level. According to the EML, these medicines are not available for use in primary healthcare. This finding showed that delays in the revision of the STG/EML resulted in the development of newer guidelines, e.g. the STI guidelines, which make recommendations not in line with the national STG/EML. Furthermore, frontline healthcare providers, particularly in clinics and health centres aired their frustration on certain medicines that were, according to the EML, not available at these levels of care yet they see a lot of patients who need such medicines. Health centre level frontline providers questioned restriction of some medicines to hospital level, yet there were doctors in health centres who could manage patients the same way they would be managed in hospitals.

> "I want to believe that the ministry of health has got the presentation on power point made by Facility Y in which we are asking the ministry to revise some parts that are in the guideline to make more drugs available for health centres and clinics. We know that in Swaziland there are drugs that you can't find in clinics. There are drugs that you can't find in health centre though those drugs are available in the country (FM_MO_Sec_L)."

In such cases frontline healthcare providers from clinics and health centres mentioned that they referred patients to hospitals and it is often geographically difficult for patients to access these. Most medicines for managing NCDs were reported to not be allocated for use in primary healthcare facilities in the STG and hence not available at primary level facilities. The following are medicines for management of NCDs available at primary level of care: for hypertension—only hydrochlorthiazide; for diabetes—none of the medicines are available at primary level; for arthritis—indomethacin, colchicine, allopurinol, acetylsalicylic acid, and procaine penicillin/erythromycin [for osteomyelitis] are indicated for primary level use). Antibiotics were also reported as medicines that are not available at primary level of care; these were: penicillins (amoxicillin + clavulanic acid, flucloxacillin) all cephalosporins, sulphonamides (trimethoprim/sulphamethoxazole (400/80) injection), Macrolides /lincosamides/streptogramins (clarithromycin, clindamycin), aminoglycoside (streptomycin and vancomycin injections), quinolones (ciprofloxacin 250mg tablet, while the 500mg tablet is indicated for primary level use), and nitrofurantoin.

**Availability of essential medicines.** Frontline healthcare providers reported that essential medicines were constantly out of stock at the CMS and in health facilities. Stock-outs were reported as barriers that affect adherence to the STG/EML and rational use of medicines. Due to frequent unavailability of medicines, frontline healthcare providers reported that they often find themselves out of options on what to prescribe for the patients, and at the same time, found it difficult to send patients home with no medicines to alleviate their suffering. In such instances, frontline healthcare providers reported that they send patients to private sector facilities and sometimes prescribe out of the STG/EML and give medicines that might not be appropriate for the condition. Frontline healthcare providers stated that patients needed to have money to buy medicines and for transport to get to private sector facilities. Medicines that were reported to constantly be out-of-stock were those for managing NCDs. Often, patients would report to not have money to travel and buy medicines and such patients then discontinue taking their NCD medicines, compromising their management.

Some reasons that were cited for stock-outs of essential medicines were poor stock management practices which result in inadequate amounts ordered from the CMS.

"I think there are challenges with stock ordering from the facilities. You find that they run out because they haven't ordered the right quantities due to poor stock management practices (KI_P_C_M)."

Delayed ordering by facilities, long lead times for CMS to deliver in facilities, inconsistent/inadequate supply of medicines by the CMS, and poor communication on stock availability between the CMS and facilities were also reported to affect stock availability and ultimately RMU.

"We only hear about stock shortages from facilities when they now are complaining that they are not getting their order, so there is no transparent communication between the central medical stores and the facility or at least the region (KI_P_C_M)."

**Lack of pharmacy personnel in primary levels of care.** Frontline healthcare providers reported that clinics do not have pharmacy personnel, and this made it difficult for nurses to manage facility stocks of medicines (monitoring average monthly consumptions, quantities to order, and share slow-moving stock with other facilities to minimize expiries) as their professional qualification does not equip them to do this work. Though frontline healthcare providers reported that they receive off-and-on-site intermittent trainings on stock management from the CMS and implementing partners, they highlighted that not all nurses in a facility would receive the training. Furthermore, rotations of nurses between facilities resulted in some facilities having no nurses trained on stock management.

**Inactive PTCs.** Key informants and frontline healthcare providers reported that health centres and hospitals had PTCs, however, most were reported to not function adequately. Information gathered from some participants highlighted that monitoring of activities of PTCs was through meeting minutes that are submitted to the MoH headquarters. On probing on activities performed by PTCs, participants mentioned that meetings mainly discussed errors in prescribing; and due to prescribers feeling as if they were targeted during these, they often did not attend meetings. This resulted in committee members not forming a quorum and hence cancelling most meetings, ultimately rendering PTCs inactive in facilities. Clinics reported that they have monthly/weekly meetings which they used to discuss all issues pertaining their facilities—including prescribing patterns.

## Patient factors (meaning)

This section reports on the effects of cultural conventions, expectations, and socially shared meaning on rational use of medicines. Themes reported to affect prescribing were mainly patient factors. Frontline healthcare providers reported that patients took too long to present to facilities. By the time they did, ailments would have progressed to complications, making it difficult for providers to treat these patients according to the STG/EML. Frontline providers also reported that patients demanded more medicines (an issue resulting in polypharmacy) and were not comfortable to leave the healthcare facility with no or few medicines.

"There also is also pressure from the patient that require a lot of variety of medicine. Patients are not satisfied if they are going away with maybe three types of medicines when they leave the facility, ah maybe they have a flu or headache or maybe a simple injury, you

discover that they are not satisfied once they give, if you just give them one or two medicines, they want many (FM_PT_Sec_S)."

To increase the number of medicines, frontline providers mentioned that prescribers end up prescribing medicines with no or little proven effectiveness. Frontline providers also mentioned that they sometimes prescribe and dispense a lot of medication to try and cover all ailments that the patients could be suffering from.

"uhm, yah we can benefit but once we had, we want to give the best care to our clients, we want to give extra actually if I can say so we don't want the client to come back with such a case that you gave me this drug what, what, what (FM_N_Prim_L)".

On probing, the medicines that were reported to be mostly prescribed to increase medicine numbers were multivitamins, vitamin B complex, methyl salicylate and low doses of calcium gluconate. Key informants highlighted that polypharmacy was not just as a result of patients demanding many medicines from healthcare providers. They reported that polypharmacy for NCDs was very high, due to poor consulting practices by some healthcare providers.

"When it comes to chronic medications—that is where we have observed the worst case the most because you find that the prescriber, before even the chronic patient comes in for their diabetes or hypertension, the prescriber has started writing something on their prescription book the first two being a pain killer or even some multivite without assessing the patient and not knowing what is wrong with them. I mean does it mean everyone who leaves the hospital must have a pain killer, does it mean everyone is in pain—so that's why we say we have picked up elements of irrational use which we need to tell people to solve. . . Evidence of that is that at times when they [patients] are given some of these [medicines], because they know their core medicines that they need to take, and then the others [medicines] they will tell you "oh I still have that at home", so it shows that they are keeping it they're not taking it they are just taking their BP medication because that is their basic medication so they know that the others are just for pain if I may call them that [. . .Laughs. . .] (KI_P_N_H)."

Possible reasons for polypharmacy as stated by one frontline healthcare provider were poor adherence to the STG/EML, ignorance and poor knowledge

"I think it's ignorance, lack of knowledge, not following the standard treatment guideline or even trying to please the patient (FM_P_Sec_H)".

Language was also cited as a patient factor that affected rational use of medicines. In facilities that were close to the border, frontline providers mentioned that they had patients that came to access healthcare from Mozambique and it was difficult to communicate with them. Also, providers reported that the elderly often struggled to understand instructions on how to use the medicine they were prescribed which left frontline providers uncertain if such patients used the medicines correctly.

## Provider factors (competence)

Provider-oriented factors that were reported to affect rational prescribing were: symptomatic prescribing, high patient volumes, competence, individual prescriber practice ethic, poor teamwork in patient management, and poor documentation practices. Frontline healthcare

providers reported that irrational use of medicines is sometimes driven by symptomatic prescribing; whereby prescribers treat symptoms and not the diagnosis.

> "The treatment is according to symptoms, so if you are sneezing you get items for sneezing, if you are coughing you get something for coughing and if you have an itchy eye you get something for the itchy eye. Itchy nose you get something for the itchy nose, if you have irritating throat, they give you something for irritating throat, list continuous like that, instead of making a proper diagnosis: what is the proper diagnosis, maybe the person has got one diagnosis which is flu, or which is respiratory tract infection. Just treat the disease, what do they do, treat each symptom with each medication (FM_PT_Sec_L)."

Regarding patient volumes, frontline healthcare providers reported that they sometimes prescribe as many medicines as possible to cover symptoms reported by patients and ensure that they do not come back to the facility the following day and increase patient numbers.

In this study, healthcare provider knowledge and embodied skills, as highlighted by the SPT, were reported to affect prescribing practices. Health system actors reported that actors at all levels of care in the country were poorly trained on rational use of medicines, and this resulted in inappropriate use of medicines in health facilities. Though some RMU training for frontline healthcare providers was happening, key informants reported that this training was not streamlined and policy makers were not privy to information on who has and has not been trained as they were not receiving training reports from facilitators.

Frontline healthcare providers reported that Eswatini did not have pre-service orientation of foreign-qualified healthcare professionals on RMU and use of the STG/EML.

> "Their competence is not tested, in terms of prescribing or their knowledge on the disease patterns that are affecting the Southern region as well as the patterns again that affect the local region. They struggle in the early days to uh to understand the disease pattern that we do have so at the end of the day they just prescribe anything maybe according to whatever they have been trained, so there is also a general problem of not following the treatment guideline of which they are there (FM_PT_Sec_H)".

Different practice ethics by prescribers were reported to also affect rational use of medicines in Eswatini. One key informant mentioned that some prescribers were not comfortable to consult guidelines in front of patients.

> "Because people think that [. . .pause. . .] I don't know it has not been proven so I want to believe it's a perception of the health care workers because they tend to think that if they flip through their STG the patient will think they do not know, each time they are treating them they have to be checking but maybe with a mobile app on their smart phone the fears will be allayed (KI_P_IP_H)."

On probing to find how the issue of frontline healthcare providers not being comfortable to consult guidelines in front of patients could be solved, health system actors suggested the use of a mobile application.

Poor teamwork in the management of patients was also reported to result in irrational use of medicines. Pharmacy staff mentioned that prescribers were not comfortable with suggestions made by pharmacy personnel on adjusting prescriptions to meet the patients' needs and often felt undermined.

"The pharmacy personnel call that prescriber when they receive a prescription and feel there is some irrational use of medicines to discuss the issues. However, a lot of times the pharmacy staff come across challenges when they do this as some of the prescribers then feel undermined, offended, and most of the time they refuse to change the prescription (FM_PT_Sec_H)."

If pharmacy staff changed the prescription, the patient often went back to the prescriber to question the change. This caused disputes between pharmacy staff and prescribers which resulted in disciplinary meetings being called for pharmacy staff.

"It's like they really don't want to be corrected and so they called a meeting against pharmacy so our challenge is tight, but now we don't call them now you look at the prescription and you see what you can do (FM_PT_Sec_S)."

Furthermore, frontline providers reported that "being used to doing things in a certain way" resulted in inappropriate use of medicines.

"So, the weakness there is training, and not only training it's again changing the culture and the practice and culture . . . The tendency is, you see if we have a problem, we talk maybe in a therapeutic committee today and hope that people will bring sort of change, then after that the graph just goes down. People change into their old practices that's why I said it's a culture, a culture is difficult to change for someone who is, maybe, they have been taught at school that this is what they are supposed to do and so you know trying to change that culture is a long process (FM_P_Sec_M)."

Poor/no documentation of prescribing information in some facilities was reported as another factor affecting rational use of medicines. Key informants highlighted that they were not sure if facilities with no records of prescribing information did this due to lack of knowledge on the importance of documentation and record keeping. On probing as to how it was possible for a facility to not have such records, key informants reported that in such facilities the doctor/nurse writes a prescription on a notepad that the patient takes it to the pharmacy and no record of the prescription remains in the prescriber's room. Once the prescription has been filled in the pharmacy, the patient takes the original prescription with them leaving the facility with no record of what was prescribed and dispensed for the patient.

Contrary to information from key informants on record keeping, frontline healthcare providers reported that they knew and understood the importance of keeping patient records in facilities. However, the government was constantly stocked out on prescription booklets in which each prescription is triplicate. Once the prescription has been written, the first and second copies are given to the patient to take to the pharmacy for dispensing of medicine—one copy stays in the pharmacy after dispensing while one copy leaves with the patient. The third copy remains with the prescriber, hence, there then is a record of each patient that stays with the prescriber and another with the pharmacy which are kept in the facility.

Beyond identifying factors that affect rational use of medicine, health system actors had recommendations on how RMU can be promoted in the country. Such recommendations included: functional PTCs; on- and off-site training of prescribers on appropriate use of medicines and inventory management; pre-service training for healthcare providers who qualify outside Eswatini; review and update of the STG/EML; supervision and mentorship of facility staff on medicines; widening the list of medicines available at primary health care level;

development, implementation and monitoring of RMU policies; and availability of nurses trained on RMU in all facilities.

## Discussion

Findings of this study show that there is inappropriate use of medicines in Eswatini. An interaction of health system context factors, provider factors and patient factors were found to affect rational use of medicines. Reported health system factors such as poor adherence to guidelines, stock outs of essential medicines, unavailability of RMU policies, lack of RMU training for healthcare professionals, and poor functioning of PTCs negatively affected prescribing practices. Poor adherence to treatment guidelines is not peculiar to Eswatini as similar findings have been reported in Sierra Leone [34], Botswana [35], and China [18]. The importance of adhering to national treatment guidelines cannot be over emphasized as national guidelines are developed as a strategy to improve rational use of medicines.

Poor inventory management and poor communication on stock availability both resulting in stock outs of essential medicines were reported to affect rational use of medicines in the country. Unavailability of essential medicines compromises prescribing in line with national guidelines. In Eswatini, the Ministry of Health receives its financial budget from the Ministry of Finance. Unavailability of essential medicines could be due to lack of funds by the Ministry of Finance, delays in the release of the budget by the Ministry of Finance for health commodities, and poor stock management by the CMS and facilities. Negative effects on stock levels due to limited financing of health commodities from the government are not unique to Eswatini. In China, limited financing from local governments was reported to influence reduction of medicine stocks and negatively affect service delivery to local consumers [36]. To improve medicine availability and promote RMU in Eswatini, concerted efforts to ensure timely availability of funds for health commodities between the Ministries of Health and Finance, and inventory management, need to be strengthened.

Pharmaceutics and therapeutics committees were reported to be in place in health centres and hospitals. However, informants reported that these PTCs were not as active as they expected them to be. Poor performance of PTCs has also been reported in rural Thailand where they reported that, among other factors, the performance of hospital PTCs was compromised by professional and personal prejudices and conflicts, poor performance monitoring, poor communication, lack of a standardized way of selecting medicines and over-stretched committee members [37]. Efforts to ensure that PTCs, as custodians of medicines in health facilities, are actively performing their duties need to be strengthened in order to improve the rational use of medicines in Eswatini.

Prescriber-oriented factors that result in irrational use of medicines reported in this study included: the prescriber's knowledge, personal preference and experience in clinical management of patients, and prescribers not being comfortable to consult guidelines in front of patients. Irrational use of medicines due to prescriber-oriented factors is not unique to Eswatini as literature shows that insufficient prescriber knowledge, their personal beliefs, and their fear that not giving certain medicines lead to patient complications contribute to irrational prescribing of medicines [6, 11, 12, 38–41]. Availability of national policies on RMU and a standardized orientation programme for recently qualified healthcare professionals could help address these prescriber-oriented factors.

This study also found that patient factors, where patients influence prescribers to prescribe certain medicines for them, affect prescribing practices and rational use of medicines. Similar findings have been reported in Tanzania [42]. Literature further shows that in the private

sector patients are thought to negatively influence prescribers' prescribing practices since prescribers are likely to lose patients if they do not give in to their demands [13].

Engagement of stakeholders in the different Ministries responsible for availability of medicines, capacitation of healthcare professionals, up-to-date STG/EMLs, functioning PTCs, patient education, and supportive supervision on rational use of medicines could potentially improve RMU in Eswatini.

## Trustworthiness and rigour

We acknowledge that some respondents may have provided thin descriptions of prescribing practices in relation to rational use of medicines as rational medicine use was an abstract topic to them. Lincoln and Guba's criteria for generating trustworthy results were applied in conducting this study [43]. Trustworthiness and study rigour were improved by applying the following principles. The interview guides were piloted to assess the information that would likely be produced. After piloting, questions that were poorly understood by the respondent were rephrased. Other questions were added to explore other aspects that came up during piloting. The revised interview guides were then applied to conduct the main study. Also, during interviews, the PI guarded against leading participants and allowed them to freely share information. Credibility of study findings was further improved through member checks (recapping key messages emanating from the interview) at the end of each interview to check if theories formulated captured participants' views.

To ensure credibility, the audio recordings were transcribed verbatim in English. After the transcription by the research assistant, the first author rechecked to ensure that the transcriptions were accurate by re-listening to the tapes and reading the transcripts. During the entire study, a reflexive journal—forming part of an audit trail—that captured discussions and decisions of the investigators was kept. To improve dependability, two members of the research team (NBQN and LK) independently coded transcripts. Agreement was used to reach consensus. The research team had no relationship with the participants though the PI had previously worked with some of the KIs. Finally, in reporting study findings, relevant elements for reporting qualitative research (COREQ) outlined by Tong, Sainsbury, and Craig [44] were followed. Quotes obtained from the original transcripts were provided as evidence of the identified themes and subthemes.

## Study limitations

The limitation of using an exploratory study design is that it predominantly provides descriptive findings rather than explanatory (how and why things happen).

Though the SPT was a starting point in analyzing practice and helped in gaining insight into how practice can change; it has limitations in that it provides contextual analyses that cannot be generalized [45]. Hence findings from this study only apply to the facilities assessed in Eswatini. Furthermore, adapting the SPT for this study meant that we utilized it to meet our needs in order to answer questions relating to prescribing practices in Eswatini. By so doing, we could have missed other aspects that could have emanated from the data. Another limitation to this study is that most respondents were from the pharmacy cadre; hence reported findings could mainly be views from this cadre.

## Conclusion

Availability of the STG/EML to guide medicine use in public sector, not-for-profit faith-based and industrial health facilities has promoted rational use of medicines. Findings of this study highlight that in Eswatini, prescribing practices are influenced by the interaction of a

number of factors—health system, provider and patient—that span levels (facility, region, and policy-making) of the health system. Promoting rational medicines use thus goes beyond the availability of guidelines and provider training and requires concerted efforts of multiple stakeholders.

## Supporting information

**S1 Fig. Study theories and procedures.**
(DOCX)

**S1 Table. Factors affecting prescribing behaviours and how they link to the SPT.**
(DOCX)

**S2 Table. Study participants' characteristics.**
(DOCX)

**S1 File. COREQ 32 checklist.**
(PDF)

**S2 File. Key informants interview guide.**
(DOCX)

**S3 File. Frontline managers interview guide.**
(DOCX)

**S4 File. Key informants quotes.**
(DOC)

**S5 File. Frontline managers quotes.**
(DOC)

**S1 Data. Key informants study data.**
(XLSX)

**S2 Data. Frontline managers study data.**
(XLSX)

## Acknowledgments

The Institute of Tropical Medicine (ITM), financially supported the conduct of this study through the School of Public Health at the University of the Western Cape. The ICAP at Columbia University Eswatini office assisted with transport to go to health facilities and office space to work on the study. Dr Ferdinand Mukumbang provided technical support to this study. Dr Nkosinathi Ncube provided financial, moral and emotional support; Ms Sibonisiwe Ncube transcribed; and Mr Bheki Ginindza drove with me around Eswatini to show me some of the facilities.

## Author Contributions

**Conceptualization:** Nondumiso B. Q. Ncube, Helen Schneider.

**Data curation:** Nondumiso B. Q. Ncube, Lucia Knight.

**Formal analysis:** Nondumiso B. Q. Ncube.

**Investigation:** Nondumiso B. Q. Ncube.

**Methodology:** Nondumiso B. Q. Ncube.

**Project administration:** Nondumiso B. Q. Ncube.

**Resources:** Nondumiso B. Q. Ncube.

**Software:** Nondumiso B. Q. Ncube.

**Supervision:** Hazel Anne Bradley, Helen Schneider, Richard Laing.

**Validation:** Nondumiso B. Q. Ncube, Lucia Knight.

**Writing – original draft:** Nondumiso B. Q. Ncube.

**Writing – review & editing:** Lucia Knight, Hazel Anne Bradley, Helen Schneider, Richard Laing.

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
