## [Decision Letter · Decision Letter 0]

6 May 2020

PONE-D-20-01768

Health system actors’ perspectives of prescribing practices in public health facilities in Eswatini: A Qualitative Study

PLOS ONE

Dear Mrs Ncube,

Thank you for submitting your manuscript to PLOS ONE. After careful consideration, we feel that it has merit but does not fully meet PLOS ONE’s publication criteria as it currently stands. Therefore, we invite you to submit a revised version of the manuscript that addresses the points raised during the review process.

We would appreciate receiving your revised manuscript by Jun 20 2020 11:59PM. To enhance the reproducibility of your results, we recommend that if applicable you deposit your laboratory protocols in protocols.io, where a protocol can be assigned its own identifier (DOI) such that it can be cited independently in the future. For instructions see: http://journals.plos.org/plosone/s/submission-guidelines#loc-laboratory-protocols

We look forward to receiving your revised manuscript.

Kind regards,

Khin Thet Wai, MBBS, MPH, MA (Population & Family Planning Resear

Academic Editor

PLOS ONE

Additional Editor Comments:

This article addresses the factors influencing rational prescribing practice for quality medicine by application of Social Practice Theory. In general, authors should follow the submission guidelines in preparing the abstract, text structure and the citation style.

To further improve scientific integrity, authors need to pay attention to the following:

1. To add one key reference on qualitative research reporting guidelines.

2. To check and revise throughout in line with COREQ 32 item checklist for qualitative studies.

3. To reorganize the sequence of Results section in line with key themes emanated [LINES: 188-190].

2. Please ensure you have thoroughly discussed any potential limitations of this study within the Discussion section.

3. Please include additional information regarding the interview guide used in the study and ensure that you have provided sufficient details that others could replicate the analyses.

For instance, if you developed a guide as part of this study and it is not under a copyright more restrictive than CC-BY, please include a copy, in both the original language and English, as Supporting Information.

4. Please provide additional details regarding participant consent. In the ethics statement in the Methods and online submission information, please ensure that you have specified what type of consent you obtained (for instance, written or verbal, and if verbal, how it was documented and witnessed).

Reviewers' comments:

Reviewer's Responses to Questions

**Comments to the Author**

1. Is the manuscript technically sound, and do the data support the conclusions?

Reviewer #1: Yes

Reviewer #2: Yes

2. Has the statistical analysis been performed appropriately and rigorously? 

Reviewer #1: N/A

Reviewer #2: Yes

3. Have the authors made all data underlying the findings in their manuscript fully available?

Reviewer #1: No

Reviewer #2: Yes

4. Is the manuscript presented in an intelligible fashion and written in standard English?

Reviewer #1: Yes

Reviewer #2: Yes

5. Review Comments to the Author

Reviewer #1: This paper reports on qualitative research conducted in Eswatini to investigate perspectives of key health system actors on prescribing practices and factors influencing these.

I have reviewed this with a particular focus on the description of methods, and reporting of qualitative data throughout the paper. Please see feedback on recommended revisions below.

Introduction

It is not clear how the Social Practice Theory developed in this section informed the method section. It seems as you used elements of the SPT to develop your interview guide.

I would rather suggest that the authors develop a sub-section on “Conceptual framework” under the methods section.

Methods

Study design:

This should provide a clear statement on the study design, with appropriate definition and referencing. The authors state that this is an ‘qualitative exploratory study design’. This will need defining and referencing.

The authors should describe how this study is embedded in the larger intervention study, in term of research questions, methodology, ….. How are they complementary?

The sampling requires a clear summary about the people who were interviewed, especially frontline managers. How they were sampled, and how they were recruited. In addition, how the balance between urban/rural settings, public/private health facilities was ensured.

I would propose that the following information is covered:

- a sentence stating how these were sampled, including a definition of this sampling strategy, an explanation about why this sampling strategy was used, and a recognisable reference to support these ideas

- a table to clearly outline and summarize the sample, including information on number of participants, district participants, settings, level of care, owner of the facility (public/private).

- A very transparent explanation about how these participants were recruited into the study (how were they identified, how were they invited into the study, who invited them into the study etc).

- A statement on whether they were reimbursed/incentivised.

Data collection

The data collection subsection should include a clear account of the following issues:

- Why a semi-structured interview was chosen, with a reference to literature

- How long interviews lasted

- What language they were conducted in

Data Analysis

The analysis process needs to be clearly described, and referenced to a recognisable analysis process. This section lacks convincing or transparent explanation. Please clarify so there are clear statements on:

- the analysis process used, with reference to a systematic analysis process

- a step by step explanation about how the analysis was conducted, including an explanation of which analysis approach and technique (again with references) were used in this study.

Results

Lines 186-188 should do under methods section

The reference for each quote needs to be done in a systematic fashion, to provide sufficient information linking the quote back to the sample. I propose that the authors include ‘role, level of work, district interviewee is from’. These need to link back clearly to the sample section of the methods, and to the table summarizing the sample, so readers are clear who is quoted. At the moment, we have no idea how many of the interviewees are represented in the quotes selected, and how representative of the sample these selected quotes are.

Discussion

The authors should discuss the study limitation: methods and results

- The limitation is applying the Social Practice Theory

- The limitation in utilising the adapted model of practice (SPT)

- The study limitations section needs to include statements on the ‘reliability’ and ‘rigour’ of the study design. Explanations of these are given in any good methods text book (e.g. Bryman, A. (2008). Social Research Methods. Oxford: Oxford University Press)

- In the study limitations, you will also need to account for the potential issues associated with rigour and integrity associated with using a number of different topic guides throughout the study, as it ‘thins’ the data collected (i.e. not all interviews were conducted in the same manner)

Reviewer #2: This is an important study and the findings are relevant to inform interventions to improve pharmaceutical care. I would like to also congratulate the authors for this piece of work. Below are my comments and suggestions for the authors.

General comment

The manuscript is generally well organized. However, I believe some sentences can be revised to bring more clarity.

Introduction

1. The authors need to provide more context on the Standard Treatment Guidelines (STGs) and the Essential Medicines List (EML). Are the medicines in these documents assigned specific levels of care? Are there key differences between the EML and STGs? For example, do these documents differ on how they assign/restrict medicines to levels of care? These are needed to put the findings from the study in context.

Methods

1. Under study setting, it would be helpful to also indicate which cadres of health workers are available at the various levels of care.

Results

1. The authors provided the range of the number of years of experience of the study respondents. It would be helpful to also provide the median number of years of experience.

2. It would be helpful to include the breakdown of the number of respondents (especially the 32 frontline health workers) by level of care or type of facility (clinics, health centers, hospitals etc.)

3. It is hard to believe the STG/EML were mainly tailored for primary health care as reported by the authors: “current STG/EML, which were mainly tailored for primary health care” (under the section titled “availability and use of STG/EML”). This may require further explanation. Does this mean the medicines in the EML for instance, are only medicines for primary care services? The comment that “Frontline healthcare providers highlighted the difficulty for them to adhere to treatment guidelines due to restrictions on availability of medicines at certain levels of care” (under the section titled “restriction of medicines by level of care”) indicates that the guidelines may not only tailored to primary care facilities

4. This finding/sentence may also need more clarification: “However, key informants highlighted that targeting the guidelines at primary level of care resulted in secondary and tertiary level facilities prescribing outside the guidelines”. What does “targeting guidelines” mean? Does it have to do with the content of the guidelines focusing on primary care or the promotion of the guidelines to primary care facilities?

5. Line 268 -269: Please provide specific examples of NCD medicines and antibiotics that are not available at primary health care facilities.

6. PLOS authors have the option to publish the peer review history of their article (what does this mean?). If published, this will include your full peer review and any attached files.

Reviewer #1: No

Reviewer #2: No

---

## [Author Response · Author response to Decision Letter 0]

1 Jun 2020

School of Public Health

University of the Western Cape

Private Bag X17

Bellville 7535

South Africa

1 June 2020

The Editor in Chief

PLos One Journal

Dear Sir/Madam

Re: RESPONSE TO PLOS ONE REVIEWERS’ COMMENTS

The researchers wish to thank the editor and reviewers for the time availed to review the manuscript and the valuable comments offered. Below are itemized replies to the comments. The reviewers’ comments are reproduced for ease of reference, followed by the response of the researchers. Modifications are in track changes in the manuscript entitled 

“Revised Manuscript with Track Changes”.

Anonymised data sets have also been uploaded. 

Editor’s Comments

Comment: Add one key reference on qualitative research reporting guidelines.

Response: Qualitative research reporting guidelines have been referenced in the manuscript on page 25, 1st paragraph.

Comment: Check and revise throughout in line with COREQ 32 item checklist for qualitative studies.

Response: The manuscript has been revised in line with the COREQ 32 item checklist and the checklist has been uploaded as a supporting file.

Comment: Reorganize the sequence of Results section in line with key themes emanated [LINES: 188-190]

Response: The Results section has been reorganized in line with key themes – material, meaning, and competence – on pages 12-22.

Journal Requirements

Comment: Please ensure that your manuscript meets PLOS ONE's style requirements, including those for file naming.

Response: The manuscript has been revised to meet PLOS ONE's style requirements, including those for file naming.

Comment: Please ensure you have thoroughly discussed any potential limitations of this study within the Discussion section.

Response: Study limitations have been discussed on page 25 of the manuscript under the section “Study limitations”.

Comment: Please include additional information regarding the interview guide used in the study and ensure that you have provided sufficient details that others could replicate the analyses.

Response: Detail on the interview guide has been added under the “Data collection” section on pages 10-11, and under the section “Trustworthiness and rigour” on page 24; paragraph 1, lines 4-6. The interview guides (for KIs and facility managers) have also been uploaded as supporting files.

Comment: Please provide additional details regarding participant consent. In the ethics statement in the Methods and online submission information, please ensure that you have specified what type of consent you obtained (for instance, written or verbal, and if verbal, how it was documented and witnessed).

Response: Additional detail on participant consent has been added under the section “Ethical consideration” on page 11; lines 6-9.

Comment: We note that you have indicated that data from this study are available upon request. PLOS only allows data to be available upon request if there are legal or ethical restrictions on sharing data publicly.

Response: This has been rectified on the submission. For this study there are no legal/ethical restrictions on publicly sharing the data.

Comment: PLOS requires an ORCID iD for the corresponding author in Editorial Manager on papers submitted after December 6th, 2016. Please ensure that you have an ORCID iD and that it is validated in Editorial Manager.

Response: The corresponding author now has an ORCID iD.

Reviewer 1 Comments

Introduction

Comment: It is not clear how the Social Practice Theory developed in this section informed the method section. It seems as you used elements of the SPT to develop your interview guide.

I would rather suggest that the authors develop a sub-section on “Conceptual framework” under the methods section.

Response: The authors used the SPT elements to develop the interview guides for KIs and facility managers – this has been clarified under the “Conceptual Framework” section on page 8.

Methods

Comment 1: Study design: This should provide a clear statement on the study design, with appropriate definition and referencing. The authors state that this is an ‘qualitative exploratory study design’. This will need defining and referencing.

Response 1: This has been addressed under the “Study design and sampling” section on page 8; paragraph 1; lines 1-4.

Comment 2: The authors should describe how this study is embedded in the larger intervention study, in term of research questions, methodology, ….. How are they complementary?

Response 2: This has been explained under the “Study design and sampling” section on pages 8-9.

Comment 3: The sampling requires a clear summary about the people who were interviewed, especially frontline managers. How they were sampled, and how they were recruited. In addition, how the balance between urban/rural settings, public/private health facilities was ensured.

Response 3: random sampling was conducted for the big study as detailed under the “Study design and sampling” section. Frontline managers came from the randomly sampled facilities. Sampling detail has been added on pages 8-10.

I would propose that the following information is covered:

Comment 4: A sentence stating how these were sampled, including a definition of this sampling strategy, an explanation about why this sampling strategy was used, and a recognisable reference to support these ideas.

Response 4: This has been done under the “Study design and sampling” section on pages 8-10.

Comment 5: A table to clearly outline and summarize the sample, including information on number of participants, district participants, settings, level of care, owner of the facility (public/private).

Response 5: This has been clarified in the manuscript. Selection of included facilities is explained under the “Study design and sampling” section on page 8; paragraph 2; lines 5-14. Participant characteristics are further unpacked in Table 2 on pages 9-10.

Comment 6: A very transparent explanation about how these participants were recruited into the study (how were they identified, how were they invited into the study, who invited them into the study etc).

Response 6: This has been clarified on page 11, paragraph 2; lines 1-4.

Comment 7: A statement on whether they were reimbursed/incentivised.

Response 7: This has been done under the “Ethical considerations” section on page 11; paragraph 1, lines 6-7.

Data collection

The data collection subsection should include a clear account of the following issues:

Comment 1: Why a semi-structured interview was chosen, with a reference to literature

Response 1: This has been addressed under the “Data collection” section on page 10; paragraph 1, lines 8-10.

Comment 2: How long interviews lasted

Response 2: This has been addressed under the “Data collection” section on page 10; paragraph 1, lines 12-13.

Comment 3: What language they were conducted in

Response 3: This has been addressed under the “Data collection” section on page 10; paragraph 1, lines 12-13.

Data Analysis

The analysis process needs to be clearly described, and referenced to a recognisable analysis process. This section lacks convincing or transparent explanation. Please clarify so there are clear statements on:

Comment1: The analysis process used, with reference to a systematic analysis process

Response 1: This has been addressed under the “Data analyses” section on page 12; paragraph 1.

Comment 2: a step by step explanation about how the analysis was conducted, including an explanation of which analysis approach and technique (again with references) were used in this study.

Response 2: This has been addressed under the “Data analyses” section on page 12; paragraph 2.

Results

Comment 1: Lines 186-188 should do under methods section.

Response 1: Lines 186-188 have been moved to the “Methods” section under “Study design and sampling”

Comment 2: The reference for each quote needs to be done in a systematic fashion, to provide sufficient information linking the quote back to the sample. I propose that the authors include ‘role, level of work, district interviewee is from’. These need to link back clearly to the sample section of the methods, and to the table summarizing the sample, so readers are clear who is quoted. At the moment, we have no idea how many of the interviewees are represented in the quotes selected, and how representative of the sample these selected quotes are.

Response 2: This has been addressed for all quotes. Each quote is referenced by Key Informant/Facility Manager_ Cadre_Level of Care_Region

Discussion

The authors should discuss the study limitation: methods and results

Comment 1: The limitation is applying the Social Practice Theory

Response 1: This has been addressed under the “Study limitations” section on page 25.

Comment 2: The limitation in utilising the adapted model of practice (SPT)

Response 2: The authors revised the wording in the manuscript and used the SPT and not an adapted model of SPT. Limitations of using the SPT are covered under the “Study limitations” section on page 25.

Comment 3: The study limitations section needs to include statements on the ‘reliability’ and ‘rigour’ of the study design. Explanations of these are given in any good methods text book (e.g. Bryman, A. (2008). Social Research Methods. Oxford: Oxford University Press)

Response 3: This has been addressed under the “Study limitations” section on page 25.

Comment 4: In the study limitations, you will also need to account for the potential issues associated with rigour and integrity associated with using a number of different topic guides throughout the study, as it ‘thins’ the data collected (i.e. not all interviews were conducted in the same manner)

Response 4: This has been addressed under the “Trustworthiness and rigour” section on page 24; paragraph 1.

Reviewer 2 Comments

Introduction

Comment 1: The authors need to provide more context on the Standard Treatment Guidelines (STGs) and the Essential Medicines List (EML). Are the medicines in these documents assigned specific levels of care? Are there key differences between the EML and STGs? For example, do these documents differ on how they assign/restrict medicines to levels of care? These are needed to put the findings from the study in context.

Response 1: This has been addressed under the “Introduction” section on pages 5-6.

Methods

Comment 1: Under study setting, it would be helpful to also indicate which cadres of health workers are available at the various levels of care.

Response 1: This has been addressed under the “Study setting” section on page 7; paragraph 1, lines 9-15.

Results

Comment 1: The authors provided the range of the number of years of experience of the study respondents. It would be helpful to also provide the median number of years of experience.

Response1: The median number of years of experience has been added under the “Results” section on page 12; paragraph 1; line 3.

Comment 2: It would be helpful to include the breakdown of the number of respondents (especially the 32 frontline health workers) by level of care or type of facility (clinics, health centres, hospitals etc.)

Response 2: This has been addressed in Table 2 under the “Study design and sampling” on pages 9-10.

Comment 3: It is hard to believe the STG/EML were mainly tailored for primary health care as reported by the authors: “current STG/EML, which were mainly tailored for primary health care” (under the section titled “availability and use of STG/EML”). This may require further explanation. Does this mean the medicines in the EML for instance, are only medicines for primary care services? The comment that “Frontline healthcare providers highlighted the difficulty for them to adhere to treatment guidelines due to restrictions on availability of medicines at certain levels of care” (under the section titled “restriction of medicines by level of care”) indicates that the guidelines may not only tailored to primary care facilities

Response 3: The wording has been changed from “tailored to “targeted at” on page 13 under the section “Availability and use of the STG/EML” and we have further explained why the guidelines were targeted at primary health care in lines 3-8.

Comment 4: This finding/sentence may also need more clarification: “However, key informants highlighted that targeting the guidelines at primary level of care resulted in secondary and tertiary level facilities prescribing outside the guidelines”. What does “targeting guidelines” mean? Does it have to do with the content of the guidelines focusing on primary care or the promotion of the guidelines to primary care facilities?

Response 4: The wording in the section “Availability and use of the STG/EML” line 9 has been changed from “targeting the guideline” to “targeting the STG” as the STG is targeted at primary healthcare while the EML covers all levels of care.

Comment 5: Line 268 -269: Please provide specific examples of NCD medicines and antibiotics that are not available at primary health care facilities.

Response 5: Specific examples of NCD medicines and antibiotics not available at primary level of care have been provided under the section “Restrictions of medicines by level of care” on page 15; paragraph 2; lines 3-14.

Yours Faithfully,

Nondumiso BQ Ncube

---

## [Decision Letter · Decision Letter 1]

16 Jun 2020

PONE-D-20-01768R1

Health system actors’ perspectives of prescribing practices in public health facilities in Eswatini: A Qualitative Study

PLOS ONE

Dear Dr. Ncube,

Thank you for submitting your manuscript to PLOS ONE. After careful consideration, we feel that it has merit but does not fully meet PLOS ONE’s publication criteria as it currently stands. Therefore, we invite you to submit a revised version of the manuscript that addresses the points raised during the review process.Please submit your revised manuscript by Jul 31 2020 11:59PM. If you will need more time than this to complete your revisions, please reply to this message or contact the journal office at plosone@plos.org. Please include the following items when submitting your revised manuscript:

We look forward to receiving your revised manuscript.

Kind regards,

Khin Thet Wai, MBBS, MPH, MA (Population & Family Planning Resear

Academic Editor

PLOS ONE

Additional Editor Comments (if provided):

Please consider revising in accordance with the comments provided by the reviewer.

Reviewers' comments:

Reviewer's Responses to Questions

**Comments to the Author**

1. If the authors have adequately addressed your comments raised in a previous round of review and you feel that this manuscript is now acceptable for publication, you may indicate that here to bypass the “Comments to the Author” section, enter your conflict of interest statement in the “Confidential to Editor” section, and submit your "Accept" recommendation.

Reviewer #1: All comments have been addressed

Reviewer #2: (No Response)

2. Is the manuscript technically sound, and do the data support the conclusions?

Reviewer #1: Yes

Reviewer #2: Yes

3. Has the statistical analysis been performed appropriately and rigorously? 

Reviewer #1: N/A

Reviewer #2: N/A

4. Have the authors made all data underlying the findings in their manuscript fully available?

Reviewer #1: Yes

Reviewer #2: No

5. Is the manuscript presented in an intelligible fashion and written in standard English?

Reviewer #1: Yes

Reviewer #2: Yes

6. Review Comments to the Author

Reviewer #1: (No Response)

Reviewer #2: I would like to congratulate the authors for taking time to carefully incorporate the comments from the reviewers and editors. The manuscript has substantially improved. I have only a few comments below.

Introduction

-The statement “In 2017, posts for pharmacy assistants in all levels of care, but mainly clinics, were approved” is not clear. I suggest the authors expand on this statement.

Methods

-I find it confusing that the “theory informed implementation intervention (the theoretical domains framework)” was introduced in the methods section (under the section titled, conceptual framework), even though this is not the theoretical framework used for this qualitative study. I agree it will be nice to have a subsection on the conceptual framework in the methods section. However, I believe this subsection should focus on the framework that is relevant for the methodology of this particular study. This will be a good place to discuss how the framework informed the design of interview guides etc.

-Data Management: I do not believe it is necessary to disclose what the codenames of the transcripts mean. A general statement that the data were anonymized, and stored securely should suffice.

-Data analysis: It seems the information in the second paragraph should come before the first paragraph.

-Study design and sampling: I do not believe it is necessary to talk about the sampling of prescriptions which was done in the larger study but irrelevant (and perhaps distracting) for this qualitative study. Since the qualitative interviews were with health facilities, the description on how these facilities were sampled should be enough.

Results

-Availability and use of STG/EML: The authors stated, “Information from key informants highlighted that public sector prescribing practices were influenced by the current STG/EML, which were mainly targeted at primary health care (though they can still be used at secondary and tertiary levels of healthcare)”.. the authors should revise the first sentence of this section which says that both the STG and the EML are focused on primary care.

-Restriction of medicines by level of care: The authors stated, “Frontline healthcare providers highlighted the difficulty for them to adhere to treatment guidelines due to restrictions on availability of medicines at certain levels of care. An example cited was the unavailability of ceftriaxone and azithromycin at primary healthcare levels yet the latest guidelines for managing sexually transmitted infections recommend these medicines as first line therapy and for them to be available at primary healthcare level.” This is one example where referring to the STGs and the EML as if they were one document could mask important findings that may have implications for the revision of these documents. As I stated in the introduction, the EML restricts medicines to specific levels of care while the STG does not. This should be made clear in the presentation of the findings in the above sentence. If I understand correctly, based on the EML, primary care level facilities are not allowed to stock certain medicines. However, the STGs expect these facilities to treat conditions that require the use of medicines the EML restricts to higher level facilities. So there is a discordance between the EML and the STGs which may be impacting rational use of medicines.

7. PLOS authors have the option to publish the peer review history of their article (what does this mean?). If published, this will include your full peer review and any attached files.

Reviewer #1: No

Reviewer #2: No

---

## [Author Response · Author response to Decision Letter 1]

17 Jun 2020

Reviewer 1’s Comments

Introduction

Comment: The statement “In 2017, posts for pharmacy assistants in all levels of care, but mainly clinics, were approved” is not clear. I suggest the authors expand on this statement.

Response: This comment has been addressed and the sentence improved on page 4; lines 84-86 in the manuscript.

Methods

Comment: I find it confusing that the “theory informed implementation intervention (the theoretical domains framework)” was introduced in the methods section (under the section titled, conceptual framework), even though this is not the theoretical framework used for this qualitative study. I agree it will be nice to have a subsection on the conceptual framework in the methods section. However, I believe this subsection should focus on the framework that is relevant for the methodology of this particular study. This will be a good place to discuss how the framework informed the design of interview guides etc.

Response: The information and graphic on the theoretical domains framework has been deleted and the social practice theory has been used as it is relevant for this study. Information on how the SPT was used to design interview guides is available on page 7; lines 153-156.

Data Management

Comment: I do not believe it is necessary to disclose what the codenames of the transcripts mean. A general statement that the data were anonymized, and stored securely should suffice.

Response: The information has been modified on page 9; line 191.

Data analysis

Comment: It seems the information in the second paragraph should come before the first paragraph.

Response: The paragraphs have been re-arranged on page 9; paragraphs 1 and 2.

Study design and sampling 

Comment: I do not believe it is necessary to talk about the sampling of prescriptions which was done in the larger study but irrelevant (and perhaps distracting) for this qualitative study. Since the qualitative interviews were with health facilities, the description on how these facilities were sampled should be enough.

Response: The information on sampling of prescriptions has been deleted and only information on sampling of facilities left in the manuscript on page 5; lines 115-117.

Results

Comment 1: Availability and use of STG/EML: The authors stated, “Information from key informants highlighted that public sector prescribing practices were influenced by the current STG/EML, which were mainly targeted at primary health care (though they can still be used at secondary and tertiary levels of healthcare)”.. the authors should revise the first sentence of this section which says that both the STG and the EML are focused on primary care.

Response 1: The sentence has been modified on page 10; lines 232-235.

Comment 2: Restriction of medicines by level of care: The authors stated, “Frontline healthcare providers highlighted the difficulty for them to adhere to treatment guidelines due to restrictions on availability of medicines at certain levels of care. An example cited was the unavailability of ceftriaxone and azithromycin at primary healthcare levels yet the latest guidelines for managing sexually transmitted infections recommend these medicines as first line therapy and for them to be available at primary healthcare level.” This is one example where referring to the STGs and the EML as if they were one document could mask important findings that may have implications for the revision of these documents. As I stated in the introduction, the EML restricts medicines to specific levels of care while the STG does not. This should be made clear in the presentation of the findings in the above sentence. If I understand correctly, based on the EML, primary care level facilities are not allowed to stock certain medicines. However, the STGs expect these facilities to treat conditions that require the use of medicines the EML restricts to higher level facilities. So there is a discordance between the EML and the STGs which may be impacting rational use of medicines.

Response 2: Clarifications on information from respondents around the STG and EML have been made on page 10; lines 232-242.

---

## [Editor Report · Decision Letter 2]

17 Jun 2020

Health system actors’ perspectives of prescribing practices in public health facilities in Eswatini: A Qualitative Study

PONE-D-20-01768R2

Dear Dr. Ncube,

We’re pleased to inform you that your manuscript has been judged scientifically suitable for publication and will be formally accepted for publication once it meets all outstanding technical requirements.

Kind regards,

Khin Thet Wai, MBBS, MPH, MA (Population & Family Planning Resear

Academic Editor

PLOS ONE
---

## [Editor Report · Acceptance letter]

25 Jun 2020

PONE-D-20-01768R2 

Health system actors’ perspectives of prescribing practices in public health facilities in Eswatini: A Qualitative Study 

Dear Dr. Ncube:

I'm pleased to inform you that your manuscript has been deemed suitable for publication in PLOS ONE. Congratulations! Your manuscript is now with our production department. 

Kind regards, 

on behalf of

Dr. Khin Thet Wai 

Academic Editor

PLOS ONE